# Challenges of Colistin Use in ICU and Therapeutic Drug Monitoring: A Literature Review

**DOI:** 10.3390/antibiotics12030437

**Published:** 2023-02-22

**Authors:** Jitka Rychlíčková, Vendula Kubíčková, Pavel Suk, Karel Urbánek

**Affiliations:** 1International Clinical Research Center, St. Anne’s University Hospital Brno, Pekařská 664/53, 656 91 Brno, Czech Republic; 2Department of Pharmacology, Faculty of Medicine, Masaryk University, Kamenice 753/5, 625 00 Brno, Czech Republic; 3Department of Pharmacology, Faculty of Medicine and Dentistry, Palacky University in Olomouc, Hněvotínská 3, 775 15 Olomouc, Czech Republic; 4Department of Anaesthesiology and Intensive Care, St. Anne’s University Hospital Brno and Faculty of Medicine, Masaryk University, Pekařská 664/53, 656 91 Brno, Czech Republic

**Keywords:** colistin, quantification methods, drug stability, pharmacokinetics, drug monitoring

## Abstract

The emerging resistance of Gram-negative bacteria is a growing problem worldwide. Together with the financial cost, limited efficacy, and local unavailability of newer antibiotics or their combinations, it has led to the reintroduction of colistin as a therapeutic alternative. Despite its protracted development and availability on the market, there is now a complex maze of questions surrounding colistin with a more or less straightforward relationship to its safety and efficacy. This review aims to offer a way to navigate this maze. We focus on summarizing the available literature regarding the use of colistin in critically ill patients, particularly on stability, pharmacokinetics, methods for determining plasma concentrations, and therapeutic drug monitoring benefits and limitations. Based on these data, we then highlight the main gaps in the available information and help define directions for future research on this drug. The first gap is the lack of data on the stability of intravenous and nebulization solutions at clinically relevant concentrations and under external conditions corresponding to clinical practice. Furthermore, pharmacokinetic-pharmacodynamic parameters should be validated using standardized dosing, including a loading dose. Based on the pharmacokinetic data obtained, a population model for critically ill patients should be developed. Finally, the interference of colistin with extracorporeal methods should be quantified.

## 1. Introduction

Colistin (polymyxin E) is an old lipopeptide antibiotic whose isolation from the fermentation products of *Paenibacillus polymyxa* and development occurred mainly in the 1940s [1,2,3,4]. Colistin was authorized by the US FDA in 1959 [1]. Later, to reduce adverse effects, it was modified into the prodrug of colistin methanesulphonate (CMS) and successfully registered. Despite this modification, the severity of adverse effects, especially when compared to newer antibiotics, was still high, and the use of colistin was minimized in the 1980s and 1990s [5]. However, the situation in the last two decades has changed, mainly due to the emerging resistance to carbapenems, fluoroquinolones, and aminoglycosides. Another reason is the financial cost or factual local unavailability of newer antibiotics or combinations (ceftazidime/avibactam, meropenem/vaborbactam, plazomicin) which are, moreover, not clinically effective to treat carbapenem-resistant *Acinetobacter baumanii* (CRAB) infections. Therefore, colistin is currently becoming a therapeutic alternative again for infections caused by some multidrug-resistant Gram-negative pathogens [1,4,5,6,7]. Colistin is a narrow-spectrum antibiotic. In vitro, colistin shows efficacy against *Acinetobacter baumannii*, *Pseudomonas aeruginosa,* and most *Enterobacteriaceae* (*Escherichia coli, Enterobacter* spp., *Salmonella* spp., *Shigella* spp., and *Klebsiella*). Naturally resistant pathogens are *Proteus* spp., *Burkholderia pseudomallei*, *Serratia* spp., *Morganella morganii*, *Providentia* spp.; further, Gram-positive bacteria, anaerobes, an eukaryotes are typically resistant [1,2,8,9,10]. According to the Infectious Disease Society of America (IDSA) guidelines, colistin in monotherapy is considered the last choice for therapy of uncomplicated cystitis caused by carbapenem-resistant *Enterobacterales* [11]. In mild infections (e.g., tracheitis, urinary tract infections, skin and soft tissue infections) caused by CRAB, the preferred agent is ampicillin-sulbactam in monotherapy, and colistin is one of the alternate agents [12]. Combination therapy with at least two active agents, whenever possible, is suggested for the treatment of moderate to severe CRAB infections, at least until clinical improvement is observed, and polymyxins are preferred for this combination [12]. A similar recommendation is in the International Consensus Guidelines for the Optimal Use of the Polymyxins—for invasive infections due to CRAB, colistin (or polymyxin B) should be used in combination; if a second agent is unavailable, monotherapy is recommended (weak recommendation) [13]. The same guidelines suggest combination therapy, including colistin for invasive infections due to carbapenem-resistant *Pseudomonas aeruginosa* and carbapenem-resistant *Enterobacteriaceae* [13].

It is due to the replacement of colistin by newer antibiotics in the 1980s and 1990s that we are now facing a situation with limited information on its physicochemical and pharmacological properties, which are crucial for its appropriate and safe use in clinical practice, especially in critically ill patients. Such aspects are the issues of the in vitro stability of colistin and CMS and concentration-dependent tissue toxicity, methods for estimating colistin and CMS plasma concentrations, and the long-term stability of these plasma samples. Issues of a more clinical nature are the pharmacokinetic/pharmacodynamic (PK/PD) parameters of colistin and target plasma concentrations, sampling timing for the therapeutic drug monitoring (TDM), and, last but not least, the impact of extracorporeal methods used in critically ill patients on colistin and CMS pharmacokinetics (renal replacement therapy (RRT), extracorporeal membrane oxygenation (ECMO)).

## 2. Colistin Physicochemical Properties and Their Consequences

### 2.1. Colistin Chemical Profile, Mechanism of Action

Colistin is an amphiphilic lipopeptide molecule consisting of several parts—a cyclic heptapeptide, forming the hydrophilic part, linked by a tripeptide bridge with a fatty acid, representing the hydrophobic part (see Figure 1) [14]. Five diaminobutyric acid residues are attached to this core structure, which represent free amines that give the molecule a positive charge at physiological pH. The attachment of methane-sulfonate residues by covalent bonds to the five diaminobutyrate residues of the colistin molecule results in CMS, a negatively charged antimicrobially completely inactive prodrug developed to reduce colistin toxicity [15].

The positive charge of the colistin molecule determines one of its putative mechanisms of action—the competitive dislocation of calcium and magnesium ions from their binding to phosphates. The subsequent binding of colistin to these negatively charged components of the lipopolysaccharide supports the formation of pore-like structures [2,4,16]. Colistin molecule destabilizes the three-dimensional structure of the outer membrane of Gram-negative pathogens, and colistin (or other antibiotics) is allowed to reach the level of the inner membrane [1,2,4,16,17,18,19]. Here, again, the detergent effect of colistin becomes apparent; the change in the integrity of the inner membrane leads to the loss of its function, leakage of intracellular contents, and cell lysis. An alternative membrane-related mechanism of action of polymyxins refers to the theory of phospholipid cross-linking and exchange between the outer and inner membranes with subsequent induced osmotic imbalance and cell lysis [19]. Secondary mechanisms include the so-called anti-endotoxin effect, where colistin scavenges and neutralizes endotoxins [2], the noncompetitive inhibition of bacterial respiratory chain enzymes localized on the inner membrane [4,20], hydroxyl radical production via the Fenton reaction, and oxidative damage to DNA, lipids, and proteins [21]. A comprehensive diagram of the above mechanisms of action of colistin is offered by El-Sayed Ahmed et al. [22]

Both colistin and CMS are not chemically clearly definable molecules. Colistin is produced by fermentation, so it is a mixture of more than thirty components that are not in a constant ratio. For this reason, the molecular weight of colistin is not precisely determined, and its quantity is expressed in both conventional SI units and international units (IU). The main components are colistin A and colistin B, whose molecular weights are 1169.5 and 1155.4, respectively [17,23,24]. Similarly, the derivatization of the diaminobutyrate residues of colistin can lead to the attachment of two or, in contrast, no methane-sulfonate groups to individual primary amines [25]. Thus, CMS is a complex mixture of methane-sulfonated derivates, mainly CMS A and CMS B [17].

The reverse conversion of CMS to colistin occurs spontaneously in aqueous solutions in vivo and in vitro. CMS A and CMS B, as well as other derivates present, hydrolyze to a series of partially methane-sulfonated derivates and colistin [17]. After the cleavage of all sulphomethyl units, when the molecule becomes more positively charged with each cleaved sulphomethyl substituent, their antimicrobial activity is restored. Partially sulphomethylated derivatives lack antimicrobial activity.

### 2.2. Bottlenecks in Analytical Methods for Colistin Quantification

The variability in the chemical structure and antimicrobial effect of individual derivates and the presence of an unstable, spontaneously hydrolyzing prodrug complicates the development of a reliable method for quantifying colistin and CMS in a given matrix, whether it is a patient blood sample for TDM or a sample of an infusion or nebulization solution to determine stability and storage conditions in vitro.

Microbiological methods are simple and inexpensive, but time-consuming and inaccurate. They are also not suitable for determining colistin from biological samples concurrently containing CMS. The latter will be hydrolyzed to colistin during the long incubation period, thus giving falsely high concentrations [18]. Issues related to sensitivity and selectivity are also encountered with immunological methods, methods using thin-layer chromatography or capillary electrophoresis [26]. The use of high-performance liquid chromatography (HPLC) with different detection methods seems to be more appropriate. Since colistin exhibits poor absorption of ultraviolet radiation and shows no native fluorescence, it is essential to use a derivatization procedure with UV-absorbing or fluorescent reagents. Although fluorescence detection after derivatization is a sensitive method, the disadvantages listed below make it rather an alternative method. Limitations in repeatability in the determination of lower colistin concentrations, time demands, and the complexity of sample preparation are particularly troublesome. The optimal analytical method should be fast, simple, accurate, and sufficiently sensitive. These conditions are currently best fulfilled by the liquid chromatography with mass detection (LC/MS), as evidenced by recently published methodologies [26,27]. The LC/MS provides high selectivity, sensitivity, does not require prior derivatization, and analyses are performed in minutes.

In the determination of colistin concentrations, the two main components, colistin A and colistin B, are measured; CMS is determined indirectly through acid hydrolysis of the respective samples and subsequent determination of colistin due to the instability of the CMS molecule [1,17,26]. Most methods were carried out on C18 analytical columns, using water with acetonitrile or methanol with 0.1% formic acid as the mobile phase. The acidification of the mobile phase is suitable for the formation of doubly or triply charged ions. Polymyxin B or B1 were used as internal standards. The most commonly reported MS detector is the triple quadrupole. Electrospray ionization (ESI) then generates multiply charged ions produced by protonating or deprotonating free amine groups of colistin. The choice of ion to be selected as a precursor depends on the stability, intensity, and signal-to-noise ratio of the corresponding ions. For better spectrometer response, colistin is more often measured in the ESI positive mode with *m*/*z* [M + H]^+^, [M + 2H]^2+^ and [M + 3H]^3+^, at *m*/*z* 1170, 586, 391 for colistin A and 1156, 579, 386 for colistin B. The most frequently mentioned product ion corresponds to values in the range *m*/*z* 100.9–101.4 [26,27]. The product ions and a summary of the selected methods are listed in Table 1 [17,28,29,30,31,32,33,34,35,36].

Various procedures have been proposed for the pre-analytical processing of biological samples (predominantly plasma or serum): precipitation of plasma proteins with organic solvents with or without the addition of acid, precipitation of the protein followed by solid phase extraction (SPE), or SPE followed by concentrating the sample by dry evaporation and reconstitution. Although protein precipitation is an inexpensive and rapid method of sample preparation, it also poses the risk of inadequate sample clean-up and increased risk of matrix effects. Sample preparation by SPE provides a clean extract and is the choice for minimizing matrix effects. However, the more laborious and expensive sample preparation procedure can be a disadvantage [17,18,37].

In the context of guaranteeing the most accurate determination of colistin and CMS concentrations in the sample, it is worth noting the potential adsorption of colistin to some plastics, especially polystyrene or polysulphone (the latter relevant as the material of some hemodialyzer membranes); in contrast, polypropylene has a lower risk of adsorption [38,39]. This is due to the physicochemical properties of colistin, especially the amphiphilic nature and positive charge of its molecule, and the risk factors are the handling of the sample in the liquid state, the number of exposures to new, unsaturated surfaces, and the concentration of colistin [38]. However, when processing plasma or serum, the proteins contained prevent the adsorption of colistin to plastic surfaces [38].

### 2.3. Colistin Stability and Related Issues for Clinical Practice

In clinical practice, colistin is now used both in the form of colistin sulphate and CMS, which spontaneously converts into the antimicrobial active colistin; the rate and extent of this transformation depend on a range of external conditions and, of course, it occurs in vitro. One of these external conditions is the concentration of CMS in the solution. The amphiphilic nature of both colistin and CMS molecules determines the ability to form micelles and colloidal aggregates, enhancing stability with increasing concentration [40]. The compositions and pH of the carrier solution, the ambient temperature, and the material of the tubes/infusion containers further influence the stability of both molecules.

The available stability data can generally be divided into three principal categories, taking into account the above factors affecting stability and reflecting clinical situations—nebulization solutions (tested CMS concentrations 75–77.5 mg/mL), solutions for intravenous administration (tested concentrations of CMS 0.8–4 mg/mL), and plasma samples for TDM (tested concentrations 2 mg/L and 30 mg/L of CMS, 1.7 mg/L of colistin and 0.05–9.0 mg/L of colistin A and colistin B, respectively), including freeze–thaw stability [17,32,41,42,43,44,45,46].

The stability of the CMS solution for nebulization has been addressed in two papers following the FDA recommendations conditioned by the case of acute respiratory distress syndrome after the administration of CMS premix [41,42,47]. Both studies used virtually comparable concentrations of CMS solution—77.5 mg/mL in sterile water and normal saline (NS) [41] and 75 mg/mL of CMS in sterile water, respectively [42]. Both studies demonstrated excellent stability of CMS solutions with less than 1% formed colistin throughout the monitored period—24 h at 21 °C [42] and 1 year at 4 °C and 25 °C in the dark [41].

Wallace et al. tested the stability of CMS at a concentration for intravenous administration, specifically 4 mg/mL in both NS and 5% glucose in an infusion bag stored in the dark at 4 °C and 21 °C. There was a gradual increase in colistin concentrations in both carrier solutions; within 48 h, approximately 4% of CMS spontaneously converted at the higher temperature, whereas only 0.3% did so at the lower temperature [41]. Within 12–24 h after reconstitution, times that replicate the potential length of a single infusion in continuous administration, approximately 2–3% degraded to colistin at room temperature [41]. Abdulla et al. and Post et al. studied the stability of CMS in elastomeric pumps, i.e., in a different material and simultaneously at lower concentrations—0.8 mg/mL [43], and 0.8 mg/mL, 1.6 mg/mL, and 2.4 mg/mL, respectively [44]. A relationship between stability and concentration was proven—the most significant CMS conversion occurred at the lowest concentration (3.7% of CMS hydrolyzed during 8 days of storage at 20 °C and in the light vs. 2.6% and 2.3% at higher concentrations) [44]. CMS stability in the infusion bag (2 million IU of CMS in 100 mL NS; CMS concentration 1.6 mg/mL) during the 8 days was better in comparison to all concentrations used in the elastomeric pumps (the proportion of formed colistin was 1.7% and 2.1% at 4 °C and 20 °C, respectively) [44]. CMS concentration in intravenous solutions used in clinical practice typically ranges between 4.8–14.4 mg/L.

The stability of CMS and colistin at clinically relevant plasma concentrations was addressed by Dudhani et al. In their experiment, three types of samples were prepared—colistin 1.7 mg/L and CMS 2 mg/L and 30 mg/L in plasma at pH 7.4; the samples were further stored at −20 °C, −70 °C, and −80 °C [45]. The CMS concentration in both types of samples remained stable at lower temperatures for 4 months; however, at −20 °C, there was significant degradation to colistin, with a decrease of more than 26% in CMS (initial concentration 2 mg/L) after two months of storage and a concomitant measurable level of de novo formed colistin (approximately 0.4 mg/L); the stability of CMS at higher concentrations was better. Concerning the stability of colistin, there is again an explicit temperature dependence: at −70 °C and −80 °C, the degradation of colistin did not exceed 7% for 6–8 months, whereas at −20 °C, a similar extent of degradation was observed after only one month [45]. Similar data have been published by Gobin et al. [17]. Regarding the stability of CMS and colistin in the case of thawing and refreezing of the plasma sample, the available data show good stability with two [17] or three (colistin only) freeze/thaw cycles, respectively [32,46]. Considering pH as another condition affecting the stability of CMS and colistin in samples, it may be reasonable to take the acid–base balance into account in colistin TDM, especially in critically ill patients, with these data usually available [48].

### 2.4. Other Colistin Solution Properties and Their Consequences

The amphiphilic nature of the CMS molecule and the concentration-dependent ability to form micelles and colloidal aggregates determine not only the stability of the solution but also changes in its viscosity. This plays an important role, especially concerning aerosol formation in inhalation therapy. This issue was addressed by Bihan et al. [49]. The authors compared a standard dilution of 4 million IU CMS in 12 mL NS (26 mg/mL) and an experimental dilution of 4 million IU CMS in 6 mL NS (53 mg/mL), both in terms of the nature of the particles produced and the changes in pharmacokinetics. With the more concentrated solution, a larger size of particles generated was initially observed, but still within the limit that allows deposition in the distal airways. The total duration of nebulization was significantly shorter with the more concentrated solution. In contrast, the pharmacokinetics of CMS and colistin did not change with dilution [49].

Another important physicochemical property in the context of inhalation administration is osmolality, but this depends more on the type of carrier solution used for CMS. Dodd et al. investigated objectively measured changes in lung function and subjective sensations associated with the inhaled administration of hypotonic (sterile water as the carrier solution), isotonic (water and NS 1:1), and hypertonic (NS) CMS solutions at a dose of 2 million IU in adult patients with cystic fibrosis [50]. After the administration of all solutions, there was a similar decrease in FEV1 (forced expiratory volume in one second), but the time at which the maximum decrease occurred differed: the fastest onset of change was observed with the hypertonic solution and the slowest with the hypotonic solution. Similarly, the hypertonic solution was the least preferred form by patients [50]. The induced bronchoconstriction was attributed to the tonicity of the nebulization solution, but we should not forget the detergent properties of colistin, which may also account for the dose-dependent degranulation of mast cells [51,52].

## 3. Colistin Pharmacokinetics in Critically Ill Adults and PK/PD Targets

As an inactive prodrug, CMS requires in vivo bioactivation to colistin, which, together with the very different physicochemical properties of the two substances, accounts for the complex pharmacokinetics. To achieve systemic therapeutic concentrations, CMS must be administered intravenously. Of course, CMS reaches its maximum plasma concentration (c_MAX_) very quickly, whereas colistin must first be formulated (time to reach maximum plasma concentration—T_MAX_—is delayed). The bioactivation of CMS to colistin is spontaneous and represents one of the non-renal CMS clearance pathways; other non-renal pathways are possible (e.g., hydrolysis of peptide bonds) [53]. In addition to this non-renal clearance, CMS is excreted by glomerular filtration and tubular secretion, and there is an inverse relationship between renal function and the rate and extent of CMS bioactivation to colistin. In other words, patients with renal insufficiency have a higher exposure to in vivo formed colistin, whereas patients with normal renal function are at risk of too rapid excretion of the prodrug before conversion [54]. Published data from critically ill patients suggest colistin T_MAX_ in the range of 1–8 h [55,56,57,58,59,60], and the extent of CMS conversion, this time in healthy volunteers (Japanese male subjects and Caucasian subjects), is estimated to be between 30 and 60% [61,62]. The variability of individual medicinal products (not only in terms of the manufacturer but also in terms of the batch) further contributes to the already significant interindividual variability [13,63,64,65]. The c_MAX_ of colistin ranged widely depending on evolving dosing regimens and the use of a loading dose, creatinine clearance (CrCl), or the use of renal replacement therapy (RRT) (for details, see Table 2). In general, the c_MAX_ of colistin ranged widely between 0.6 and 13 mg/L [55,56,57,58,59,60,66].

Approximately 50% of colistin binds to plasma proteins, mainly α-1-acid glycoprotein [13,72]. Higher levels of protein binding (59–74%) have been observed in critically ill patients [58]. The volume of distribution in healthy volunteers is low (14 and 12.4 L for CMS and colistin, respectively) and corresponds approximately to the volume of extracellular fluid [61]. The metabolic and elimination pathways of colistin are not fully described. Given the peptide nature, proteolytic degradation is possible, but the enzymes involved and localization are still unknown [53]. Renal clearance of colistin is very low in healthy volunteers (1.9 mL/min) due to significant tubular reabsorption (organic cation transporters OCTN1, peptide transporters PEPT2, and a low-density lipoprotein receptor megalin may be involved) [53,61]. This and the adsorption of colistin to the polysulfone membrane of the hemofilter are the reasons for the administration of higher doses of CMS in patients on continuous RRT than in patients with normal renal function [39,67]. The biological half-life is again variable, approximately 3–5 h in healthy volunteers (Japanese male subjects and Caucasian subjects) [61,62], and ranges from 3.1 to 18.5 h in the critically ill population without RRT [55,56,57,58,59].

In terms of PK/PD targets, colistin is among the antibiotics with an exposure-dependent effect; the area under the plasma concentration curve of colistin or its free fraction to MIC ratio (fAUC/MIC) correlates best with efficacy in in vitro and animal models [13]. In therapeutic use, the target AUC over a 24 h interval and at the steady state (AUC_SS,24h_) is approximately 50 mg × h/L, corresponding to a mean steady-state plasma concentration (c_SS,AVG_) of 2 mg/L [13]. Both refer to monotherapy and the total plasma concentration of colistin. Plasma protein binding, as well as inaccuracies in the determination of MICs and differences between in vitro and in vivo efficacy, are reflected in this target.

Recommended dosing in critically ill patients includes a loading dose of 9 million IU of CMS administered over 30–60 min, followed by a first maintenance dose 12 h later. The daily maintenance dose in patients with normal renal function (CrCl >70 mL/min) should be between 9 and 10.9 million IU, divided into two doses; the dose should be adjusted when creatinine clearance decreases [13,67,73]. The loading dose primarily reflects the prolonged T_MAX_ in critically ill patients and the variable extent of conversion of CMS to colistin and provides the achievement of therapeutically effective concentrations as soon as possible. The relationship between the loading dose and the risk of acute renal failure as a manifestation of nephrotoxicity is unclear, as is the appropriate loading dose in specific populations (obese patients, patients with renal insufficiency, augmented renal clearance, patients on RRT) [74]. No pharmacokinetic drug–drug interactions have been described that would require a change in the dosage of CMS. Pharmacodynamic interactions with other nephrotoxic (see Section 4.2) or neurotoxic drugs (including non-depolarizing neuromuscular blocking agents, aminoglycosides) may be clinically relevant [75,76]. A synergistic antibiotic activity with colistin combined with rifampicin or carbapenems has been shown in vitro, but the superiority of such a combination has not been demonstrated clinically [77].

Due to the poor colistin penetration into lung tissue, it was believed that nebulization would produce high drug concentrations at the site of infection. When CMS is nebulized in critically ill patients, local concentrations of colistin vary over an extensive range; however, in general, a higher percentage of CMS is converted to colistin, and the local concentrations achieved are higher than plasma concentrations during intravenous treatment [51,78]. On the other hand, there are conflicting findings on the clinical efficacy of nebulized antibiotics, which may be related to the limited number of randomized clinical trials, miscellaneous pathogens, various aerosol generation devices and methods, ventilation regimens, and questionable colistin efficacy due to potential binding to mucin [11,12,13,77]. Despite the limited permeability of CMS and colistin through barriers, systemic absorption of both drugs occurs, and systemic exposure to colistin reaches a measurable level or even observable toxicity [51,77,78]. There are inconsistent recommendations on the use and dosage of inhaled colistin as an adjuvant therapy to intravenous therapy in patients with ventilator-associated pneumonia caused by multidrug-resistant pathogens [12,13,79,80]. Namely, IDSA guidelines do not suggest adding nebulized antibiotics to treat respiratory infections caused by difficult-to-treat *Pseudomonas aeruginosa* and CRAB [11,12]. Rello et al. released a similar recommendation suggesting avoiding the use of nebulized antibiotics (including colistin) instead of the intravenous or as-added to conventional intravenous therapy already including colistin or aminoglycosides for the treatment of ventilator-associated pneumonia caused by both susceptible and resistant pathogens [80]. On the other hand, the International Consensus Guidelines for the Optimal Use of the Polymyxins recommends adjunctive polymyxin aerosol therapy for hospital-acquired or ventilator-associated pneumonia, as the committee believes that the potential benefits outweigh the risks [13]. The standard dose is 1–2 million IU every 8–12 h; however, the administration of substantially higher doses (e.g., 5 million IU every 8 h) has also been proposed [73,77].

## 4. Colistin TDM, Benefits and Limits of Routine Use in Clinical Practice

There remains space for the use of colistin in clinical practice as it represents one of the few remaining therapeutic options still effective against carbapenem-resistant pathogens (see current guidelines above) [11,12,13]. Thus, the emergence of colistin resistance poses a substantial public health risk. Recently, some reviews were published on the global prevalence of colistin resistance in CRAB [81] and *Klebsiella pneumoniae* [82]. On the other hand, the distribution of colistin resistance is difficult to assess due to methodologically challenging and limited local susceptibility testing [83]. In this context, it is worth mentioning the Carbapenem and/or Colistin-Resistant *Enterobacterales* (CCRE) survey initiative (as part of the EURGen-Net), which will provide updated information on the distribution of carbapenemase-producing *Enterobacterales* and help to better understand the capacity for colistin susceptibility testing in Europe [83,84]. Regarding the susceptibility of selected pathogens to colistin, the European Committee on Antimicrobial Susceptibility Testing (EUCAST) set the susceptibility limit for *Pseudomonas aeruginosa* at 4 mg/L and for *Acinetobacter baumannii* 2 mg/L [85]. These values enable simultaneously safe and effective therapy to be achieved. Unfortunately, we are still balancing the risk of underdosing and the development of resistance on the one hand and toxicity on the other. The use of TDM seems to be the way forward. This is confirmed by guidelines that identify the research on the optimal approach to implementing TDM as a future need [13].

TDM is a useful tool for guaranteeing drug safety and efficacy, especially for agents with a narrow therapeutic window and the expected correlation between efficacy or safety and plasma concentration. Colistin fulfills these conditions, and its physicochemical properties allow us to anticipate changes in plasma concentrations in critically ill patients alone or with concomitant extracorporeal methods such as RRT and ECMO [86]. There are only a few papers available on the real impact of extracorporeal methods on colistin pharmacokinetics. CMS and colistin can be effectively cleared by continuous RRTs (CRRT). The actual extent of extracorporeal clearance depends on CRRT type and settings, the CRRT dose administered, and the presence of residual renal function; another essential factor is colistin adsorption on the hemofilter membrane [39,65,68,69,87]. Conflicting data on the adsorption of colistin to the membrane based on prefilter and postfilter colistin values were provided only by Menna et al. (postfilter colistin was more than 70% of prefilter colistin) [66]. Only two papers have addressed the interference of colistin with ECMO—colistin sulphate was used in one publication and CMS in the other. In neither case was significant sequestration of colistin on the ECMO circuit observed [70,88].

### 4.1. TDM as an Approach to Ensure Colistin Safety

Concerning efficacy, the development of multidrug resistance is becoming one of the major clinical problems in nosocomial infections, especially in critically ill patients. Another issue is the difficulty of achieving adequate colistin plasma concentrations during CMS monotherapy, particularly in patients with good to augmented renal clearance and/or organisms with MIC greater than 1.0 μg/mL [39]. Colistin’s pharmacodynamic effect depends on the AUC to MIC ratio (AUC_SS,24h_ of approximately 50 mg × h/L is the current target) [13]. For a good AUC estimation in a population with high interindividual variability typical for critically ill patients and product variability, every pharmacokinetic algorithm or simplified approach is burdened with a significant error, and, thus, multiple sampling and individual curve construction seem to be the most accurate approach [48,71]. In the available studies of critically ill patients, 6–9 samples were drawn in a single dosing cycle, whereas a single study carried only four samples (see Table 2) [55,56,57,58,59,68,71]. Thus, the number of samples seems to be another barrier to implementing TDM in routine practice. Kim et al. suggested trough level and a blood sample collected 2 h after starting a 30 min infusion for a limited sampling strategy because this combination showed the highest correlation with AUC [71]. A logical solution to minimize the sampling required for AUC estimation is the administration of colistin (CMS) as a continuous infusion. Although this regimen would theoretically correlate well with the defined PK/PD target, this approach does not have sufficient support in the literature [89,90]. In a limited cohort, continuous infusions had no protective effect on nephrotoxicity or improvement in therapeutic effect [90].

Another limitation of colistin TDM is the issue of its actual target concentration. Although the PK/PD parameter is defined, most of the available data are insufficient to fully confirm the relationship between plasma levels and the efficacy and safety of colistin treatment [86].

### 4.2. TDM as an Approach to Reduce Colistin Toxicity

The main expected adverse effects of colistin are nephrotoxicity and neurotoxicity. In individual studies involving critically ill patients, the incidence of acute renal failure ranged between 21 and 76% and 20 and 50%, respectively [13,74]. A recent meta-analysis including only randomized trials, in which colistin was administered at a controlled dosage (mainly including a loading dose), showed the incidence of colistin-induced nephrotoxicity in critically ill patients to be 36%, with a relative risk 140% higher compared to β-lactams [91]. Risk factors for toxicity include older age, preexisting renal impairment, hypoalbuminemia, higher disease severity, concomitant therapy with other nephrotoxic drugs (vancomycin, aminoglycosides, furosemide, calcineurin inhibitors) or vasopressors, colistin dosage, and length of therapy [13]. Sorlí et al. reported that trough concentrations (c_MIN_) of colistin above 3.33 and 2.42 mg/L on day 7 and at the end of therapy, respectively, are risk factors for nephrotoxicity [92]. Shields et al. stated CMS doses higher than 5 mg/kg/day within 7 days of therapy as a risk factor [74]. A dose dependence was also observed for neurotoxicity [75]. The aim is to identify not only clinical but also genetic predictors of nephrotoxicity [93]. However, none of these are currently screened for in clinical practice.

Taking all this together, colistin is a complicated substrate with many obstacles to TDM in routine clinical practice—namely, the availability of an appropriate analytical method, spontaneous degradation of CMS and risk of detecting falsely high colistin concentrations, administration of an inactive prodrug with variable extent and rate of conversion, considerable interindividual variability, product specificity, limited reliability of pharmacokinetic modeling, or the need for multiple sampling, missing clinical confirmation of current PK/PD targets. On the other hand, routine colistin TDM can produce undeniable safety and efficacy benefits and may provide refinement of dosing strategies in specific patient populations [48,68,94,95,96].

## 5. Discussion

Colistin is returning to therapeutic use nowadays, although it is usually the last choice in treating multidrug-resistant Gram-negative pathogens. The typical target population is critically ill patients. Even after almost 80 years of colistin development and more than 60 years since its marketing authorization, we still have limited data on its stability, safety, and efficacy.

The first contributor to this lack of information is the limited use of colistin in the 1980s and 1990s [5]. Another factor is the predominant use of colistin in the form of an antimicrobial inactive prodrug, which is characterized by heterogeneity of components in the individual products evolving over time [13,63,64,65], the interindividually different extent and rate of activation to colistin and different pharmacokinetics in general [54,55,56,57,58,59,60,61,62], but also the presence of spontaneous conversion to colistin ex vivo. Thus, the physicochemical properties of CMS and colistin become another source of uncertainty with a limited degree of possible extrapolation in several preclinical and clinical questions, such as the stability of infusion and nebulization solutions under different conditions, the stability of samples for TDM, the MIC determination of pathogens in the microbiology laboratory, analytical methods for quantifying CMS and colistin in a specific matrix, the nature of particles formed during nebulization, and the rate of nebulization [17,18,26,27,32,37,38,40,41,42,43,44,45,46,48,49].

Regarding the stability of CMS and colistin in solution or other matrices, we have data for a considerably wide range of concentrations. Unfortunately, these concentrations do not fully reflect those achieved in routine clinical practice, and the transferability of these results is limited, as described above. For example, the concentration of nebulization solutions tested is approximately three times higher than that used in standard practice (1–2 million IU of CMS diluted in 3 mL or 6 mL of saline) [41,42,73]. Although there are studies proposing higher doses of CMS (4 million IU every 8 h (dilution unspecified), 5 million IU every 8 h, i.e., 40 mg/mL CMS in water for injection), the concentration is still well below the tested limit [41,42,77,97]. A similar situation applies to intravenous solutions; available stability data do not adequately reflect the variability in concentrations, infusion container materials, and environmental factors accompanying CMS administration in clinical practice. When testing the stability of these solutions, the potential adsorption of colistin to plastic surfaces must also be taken into account. The protective effect of proteins is entirely absent in these solutions [38].

The determination of MIC becomes a particular issue in the context of the colistin adsorption ability on different surfaces. Colistin interferes mainly with polystyrene, but neither glass nor polypropylene showed zero adsorption at the concentrations used in microbiological testing [38]. Polysorbate-80 reduces adsorption to polystyrene, but EUCAST does not recommend using surfactants [98]. The second fact is that directly active colistin sulphate is used for microbial testing, whereas CMS is used in clinical practice. Unfortunately, even colistin sulfate is not clearly defined chemically; in other words, it is a mixture of chemically related substances in different ratios and with various antimicrobial activities [7].

In addition to MIC determination, adsorption can affect the concentration of colistin in samples obtained by bronchoalveolar lavage or microdialysis. This can alter the target value of the PK/PD parameter for colistin and question its validity. Moreover, the progressive evolution of dosing regimens in recent years must also be considered, and currently available pharmacokinetic data need to be correlated to the dosage used (with or without a loading dose), the route of administration, the order of the monitored interval, the characteristics and condition of the patient (including the use of extracorporeal methods), and the type of product used (see Table 2). The omission of a loading dose substantially delays the achievement of therapeutic levels and can be the reason for a suboptimal clinical response [39,57,87]. Future studies, including the pharmacokinetic profile of CMS and colistin and defined clinical markers of efficacy and safety, may clarify and update the target PK/PD parameters and confirm the relationship between colistin plasma levels and the effectiveness and safety of the therapy [86]. The key to this is the implementation of TDM into routine clinical practice, including a validated method suitable for the daily determination of such a specific substrate and taking into account all the above-mentioned limitations.

The aim of this review was to summarize the available data, especially regarding the administration of colistin to critically ill patients and TDM, to highlight the weaknesses of the available literature, and then to define directions for future research on this drug.

## 6. Conclusions

Despite its toxicity, which had temporarily minimized its use, colistin represents a valuable therapeutic alternative, particularly in the context of increasing bacterial resistance worldwide. A way to optimize the benefit–risk ratio is the introduction of TDM into routine clinical practice. However, there are numerous obstacles in this path, which are conditioned by the physicochemical properties of colistin: the administration as an inactive prodrug, substantial interindividual variability, further enhanced effects in critically ill patients. Summary and precise definitions of these barriers allow the design of more practice-relevant research and clinical trials whose results can improve the colistin profile.

## 7. Future Directions

There are several directions for further research on colistin. At a purely preclinical level, one is the verification of the stability of the intravenous solution and solution for nebulization at clinically relevant concentrations and under conditions corresponding to clinical practice (e.g., volume and type of carrier solution, CMS concentration, duration of administration, including continuous infusion). Another issue to test is compatibility with other drugs. Only with this knowledge can the condition of ‘the right dose’ be fulfilled as one of the eight rights of medication administration. The requirement for stability data is more urgent for the intravenous solution than the nebulization solution because of the contradictory guidelines on colistin nebulization, among others. Another area is mapping the current microbiological and susceptibility testing practices and monitoring colistin resistance development.

At the level of pharmacological and clinical-pharmacological research, we need to obtain more pharmacokinetic and clinical-outcome data in critically ill patients receiving colistin at a standardized dosage, including a loading dose. Such well-defined data can help confirm the currently reported PK/PD parameters and design a population model. These models can help us to estimate the plasma concentration–time curve of colistin without extensive sampling and quantify changes induced by extracorporeal methods. Lastly, confirmation or a more precise definition of colistin toxicity predictors would be helpful.

Taking all this together, colistin represents a drug that deserves attention in both preclinical and clinical research.

## Figures and Tables

**Figure 1 antibiotics-12-00437-f001:**
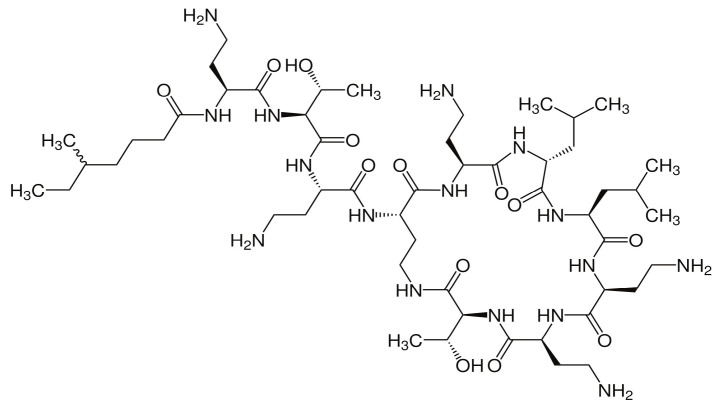
Colistin chemical structure.

**Table 1 antibiotics-12-00437-t001:** Summary of LC-MS methods for the determination of colistin and CMS.

Column Type	Internal Standard	Mobile Phase		Elution	Analysis Time [min]	ESI Mode	*m*/*z*		Ref.
		A	B				Colistin A	Colistin B	
Xbridge C18 (150 mm × 2.1 mm; 5 µm)	polymyxin B	0.1% formic acid in water	0.1% formic acid in acetonitrile	isocratic A:B (80:20)	3.8	positive	585.5 → 101.2	578.5 → 101.2	[17]
Xbridge BEH-Amide (50 mm × 2.1 mm; 2.5 µm) HILIC	not used	0.1% formic acid in acetonitrile	0.1% formic acid in water	gradient	12	positive	390.7 → 100.9	386.0 → 100.9	[28]
Synergi Fusion RP C18 (150 mm × 2 mm; 4 μm)	polymyxin B	0.1% formic acid in water	0.1% formic acid in acetonitrile	gradient	11	positive	585.7 → 101.1; 202.3; 241.3	578.7 → 101.1; 202.3; 227.3	[29]
Kinetex C18 (50 mm × 3 mm; 2.6 µm)	polymyxin B	0.1% formic acid in water	0.1% formic acid in acetonitrile	gradient	6	positive	585.5 → 101.1; 241.1	578.5 → 101.1; 227.2	[30]
Acquity UPLC BEH C18 (100 mm × 2.1 mm; 1.7 µm)	not specified	0.2% formic acid and 5% acetonitrile in water	acetonitrile	gradient	10	positive	390.60 → 101.07; 241.19	385.90 → 101.07; 227.17	[31]
Atlantis dC18 (100 mm × 2.1 mm; 3 µm)	clarithromycin	water	0.2% formic acid in acetonitrile	isocratic A:B (50:50)	4	positive	585.6 → 101.4	578.7 → 101.3	[32]
Kinetex XB-C18 (100 mm × 2.1 mm; 2.6 μm)	polymyxin B	acetonitrile:methanol (1:1)	0.1% formic acid in water	gradient	3.5	positive	390.7 → 101.3	386.0 → 101.2	[33]
MassTox	polymyxin B	0.1% formic acid in water	0.1% formic acid in methanol	gradient	3.5	positive	585.5 → 534.9; 576	578.5 → 527.9; 568.9	[34]
Acquity UPLC C18 (150 mm × 4.6 mm; 3.5 μm)	polymyxin B	0.1% formic acid in water	0.1% formic acid in acetonitrile	gradient	2.5	positive	390.9 → 385.1	386.2 → 101.0	[35]
Acquity UPLC BEH C18 (50 mm × 2.1 mm; 1.7 µm)	sulphadiazine ^13^C6	0.1% formic acid in water	0.1% formic acid in acetonitrile	gradient	3.5	positive	390.8 → 86.1; 101.0	386.0 → 86.0; 101.1	[36]
						negative	1167.8 → 1079.4; 1124.1	1153.7 → 1065.8; 1110.0	

**Table 2 antibiotics-12-00437-t002:** Characteristics of pharmacokinetic trials/case reports in adult patients receiving CMS.

Population	Number of Subjects	Renal Function	Dosage (CMS * mg Intravenous)	LD	Sampling		Pharmacokinetic Data—Colistin			Ref.
					Timing	before + after the Infusion [min]	c_MAX_ [mg/L]	T_MAX_ [min]	AUC [mg × h/L]	
critically ill	3	NS all on continuous RRT	150 q 18 h 75 q 8 h 75 q 8 h	no	day 3 day 10 day 6	0 + 10; 60; 120; 240; 360; 480; 600; before the next	2.98 2.29 2.11	10	33.6–40.8 c_SS,AVE_ 1.4–1.7 mg/L	[39]
critically ill	14	CrCl: 46.0–199.7 mL/min	225 q 8/12 h 150 q 8 h	no	after min. 2 days	0 + 10; 60; 120; 240; 360; 480	mean 2.93 SD 1.24	NS	mean 12.8 SD 5.1	[55]
critically ill	73	calculated CrCl: median 86 mL/min (range 14–368 mL/min)	LD median 160; 16 patients 600–720 MD median daily dose 480	yes	fist, random dose	during the infusion (2×) + 60–120 min; 5 min before the next	NS	NS	NS c_SS,AVE_ 3.4 mg/L **	[56]
critically ill	18	calculated CrCl: mean 82.3 mL/min (SD 24.35) mL/min	240 q 8 h or 160 q 8 h	no	first, fourth, or sixth/seventh dose	0 + 15; 30; 60; 90; 120; 240; 360; 465	*following the first dose* 0.6 ** *following next doses* 2.3 **	NS	NS	[57]
critically ill	10	calculated CrCl: 24.9–191.5 mL/min	LD 480 80–240 q 8 h	yes	first, eigth dose	0 + 15; 30; 60; 120; 240; 465	NS	NS	NS	[58]
critically ill	19	calculated CrCl: 29–220 mL/min	LD 480–720 MD 240 q 8 h or 360 q 12 h	yes	first, fifth/sixth dose	0 + 30; 60; 120; 240; 480; 720; before the next	*following the first dose* mean 2.65 range 0.95–5.1 *at steady state* range 0.68–8.72	480	NS	[59]
critically ill	2	NS both on continuous RRT	LD 720 MD 240 q 8 h	yes	first; end of second–eigth dose	0 + 30; 60; 120; 240; 360; 720; before the next	13.4 10.9	NS	216.3 179.1	[66]
critically ill	215	CrCl: median 39.8 mL/min range 0–314 mL/min 29 patients on RRT	median daily dose 480 range 180–1454	no	days 3–5	NS	NS	NS	NS median c_SS,AVE_ 2.35 mg/L; range 0.24–9.92 mg/L	[67]
critically ill	105	calculated CrCl: mean 28.7 mL/min/1.73 m^2^ range 0–169 mL/min/1.73 m^2^	median daily dose 200 range 75–410 mg	no	day 3–4	0 + end of infusion; 30; 60; 120;240; 480; 720	NS	NS	11.5–225 median c_SS,AVE_ 2.36 mg/L	[68]
critically ill	10	NS all on continuous RRT	LD 480 or 720 MD 160–240 q 8 h	yes	first, seventh dose	0 + 30; 60; 180; 360; 480	NS	NS	NS c_SS,AVE_ 4.67 mg/L	[69]
critically ill	2	CrCl: 107 mL/min 159 mL/min 156 mL/min	LD 720 MD 360 q 12 h	yes	ninth dose first dose fifth dose	0 + 60; 240; 720 120; 360; 720 120; 360; 720	NS	NS	80 61.8 56.5	[70]
NS	15	calculated CrCl: median 89.8 mL/ range 24.2–330 mL/min	mean daily dose 235.3 *** SD 79.9 ***	no	day 5	0 + 30; 60; 120; 240; 480	mean 5.50 SD 2.75	73.8 ± 69	mean 40.5 SD 20.7	[71]

* CMS 1 million IU = 80 mg of CMS; ** predicted value; *** dose as colistin base activity (mg); approximately corresponds to CMS mean 607 mg (SD 206 mg). Abbreviations: CMS—colistin methanesulphonate; c_MAX_—maximum plasma concentration; c_SS,AVE_—steady-state average concentration; CrCl—creatinine clearance; RRT—renal replacement therapy; h—hours; LD—loading dose; MD—maintenance dose; NS—not specified; q—every; SD—standard deviation; T_MAX_—time to maximum plasma concentration.

## Data Availability

Not applicable.

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
