# Peer review of "Challenges of Colistin Use in ICU and Therapeutic Drug Monitoring: A Literature Review"

_antibiotics, 2023, doi:10.3390/antibiotics12030437_

Round 1

Reviewer 1 Report

The topic of this manuscript is interesting and fits well the scope of the journal. The reviewer feels it can be accepted after some amendments.

1) The metabolic / elimination pathways of colistin and its prodrugs should be discussed. 

2) The impact of pharmacogenetics on its pharmacokinetics should be discussed.

3) Any drug-drug interaction? This issue also must be discussed.

Author Response

The topic of this manuscript is interesting and fits well the scope of the journal. The reviewer feels it can be accepted after some amendments.

First of all, we would like to thank you for your general evaluation and comments. All have been fully implemented, and we agree that the description of pharmacokinetics was incomplete in the original version of the manuscript; information on pharmacogenetics and drug interactions was missing. Thus, the inclusion of these issues has undoubtedly improved the quality of the manuscript.

1) The metabolic / elimination pathways of colistin and its prodrugs should be discussed. 

Accepted: Information on metabolism and excretion of both colistin and CMS were added to the text, or the wording was refined (see lines 272-278, 301-306 in the tracked changes version). Unfortunately, the metabolic fate of colistin is still largely unknown. Grégoire et al., 2017 proposed some steps (see citation), but no further clarification in clinical trials was published as far as we know.

No involvement of CYP450 was observed in vitro. On the other hand, MRP-mediated efflux with potential one or more uptake transporters was mentioned by BING QI, et al. (https://doi.org/10.1096/fasebj.2018.32.1_supplement.693.10). All of these are only preclinical results which is the reason why we don’t prefer to mention them in the manuscript.

2) The impact of pharmacogenetics on its pharmacokinetics should be discussed.

Accepted: Pharmacogenetic testing concerning colistin was applied to predict nephrotoxicity (here: Eadon MT, et al. Genetic Variants Contributing to Colistin Cytotoxicity: Identification of TGIF1 and HOXD10 Using a Population Genomics Approach. Int J Mol Sci. 2017;18(3):661.). The TGIF1 gene and the associated HOXD10 protein were identified as risk factors for nephrotoxicity/cytotoxicity on proximal tubule cells. However, these were preclinical tests. Because of the current lack of use of genetic testing to improve the efficacy and safety of colistin therapy, we mention this issue marginally at the end of Chapter 4.2. (see lines 441-443 in the TC version)

Unfortunately, we are unaware of any other published relationship between colistin pharmacokinetics and pharmacogenetics.

3) Any drug-drug interaction? This issue also must be discussed.

Accepted: We have added information on pharmacokinetic and pharmacodynamic drug-drug interactions at the end of the paragraph describing dosing (see lines 330-336 in the TC version). Pharmacodynamic interactions may be significant with nephrotoxic (which were already mentioned in part 4.2; the link to this part was stressed) and neurotoxic agents, which are newly listed (aminoglycosides, non-depolarizing myorelaxants).

Reviewer 2 Report

1. In the introduction, the authors mention colistin as a therapeutic option for multidrug resistant Gram-negative organisms; however, do not mention the microbiologic spectrum of activity. Moreover, the place in the therapy is not the same for all MDR organisms. This should be highlighted. 

2. The newer agents mentioned are not designed to target carbapenem-resistant Acinetobacter baumannii. This is notable as colistin is mainly used against this organism in particular. 

3. The authors highlight the many shortcomings of colistin throughout the manuscript and cite cost as a barrier to use of other agents; however the IDSA Guidance on the Treatment of Antimicrobial-Resistant Gram-Negative Infections recommends use of ampicillin-sulbactam as a first-line agent, which is not cost prohibitive. The authors should review this document with respect to additional clinical data using colistin. See: https://www.idsociety.org/practice-guideline/amr-guidance-2.0/ 

4. The manuscript perhaps spends too much time on the pharmacokinetic/pharmacodynamic profile of colistin and not enough time on the clinical utility or lack thereof. 

5. Use of nebulized colistin is problematic for several reasons in addition to PK/PD. Consider expanding on this. See aforementioned IDSA recommendations.  

6. Section 4.2 reads "TDM as an approach to ensure colistin toxicity". Please revise. 

Author Response

First of all, we would like to thank you for your comments. All have been fully implemented, and we agree that the information on guidelines and clinical use of colistin was missing. Thus, the inclusion of these issues has undoubtedly improved the quality of the manuscript.

  1. In the introduction, the authors mention colistin as a therapeutic option for multidrug resistant Gram-negative organisms; however, do not mention the microbiologic spectrum of activity. Moreover, the place in the therapy is not the same for all MDR organisms. This should be highlighted. 

Accepted: The microbiological spectrum of activity is now in the Introduction part of the manuscript. The different place of colistin in the therapy of infections caused by various pathogens was included and stressed (see lines 43-65 in the tracked changes version).

  1. The newer agents mentioned are not designed to target carbapenem-resistant Acinetobacter baumannii. This is notable as colistin is mainly used against this organism in particular. 

Accepted: This fact was explicitly stated in the Introduction (see lines 43-44 in the TC version).

  1. The authors highlight the many shortcomings of colistin throughout the manuscript and cite cost as a barrier to use of other agents; however the IDSA Guidance on the Treatment of Antimicrobial-Resistant Gram-Negative Infections recommends use of ampicillin-sulbactam as a first-line agent, which is not cost prohibitive. The authors should review this document with respect to additional clinical data using colistin. See: https://www.idsociety.org/practice-guideline/amr-guidance-2.0/

Accepted: The indications of colistin (monotherapy or combination) and its therapeutic alternatives from IDSA guidelines and International Consensus Guidelines for the Optimal Use of the Polymyxins were included in the Introduction (see lines 52-65 in the TC version)

  1. The manuscript perhaps spends too much time on the pharmacokinetic/pharmacodynamic profile of colistin and not enough time on the clinical utility or lack thereof. 

Accepted: Despite the primary aim of our manuscript - to address the main issues relevant to therapeutic drug monitoring (quantification methods, sample stability, sampling, benefits and limits for colistin TDM, including routine use in clinical practice) and related issues of drug stability and drug-material interferences - we agree that the issues of clinical nature are complementary and must be included. We have expanded the clinical information based on the reviewers' recommendations - guidelines for systemic administration (see lines 52-65 in the TC version), nebulization (see lines 343-347), drug interactions (lines 330-336), resistance issues (lines 365-382). We believe that in this way, we provide the data required by the reviewer and better balance the ratio of pharmacological and clinical data.

  1. Use of nebulized colistin is problematic for several reasons in addition to PK/PD. Consider expanding on this. See aforementioned IDSA recommendations.  

Accepted: We have added more detailed information on the inconsistency of guidelines for adjuvant inhaled administration as well as mentioned other issues related to the questionable efficacy of this procedure. (see lines 338-340, 343-347, 350,352-361 in the TC version).

  1. Section 4.2 reads "TDM as an approach to ensure colistin toxicity". Please revise. 

Accepted: The heading was revised as follows „TDM as an approach to reduce colistin toxicity“ (see line 427 in the TC version).

Reviewer 3 Report

The current study reviews the therapeutic drug monitoring benefits and limitations of colistin use in ICU. The authors shed the light on the chemical structure, analysis and pharmacokinetics of colistin providing its benefits and toxicity. The topic is suitable and interesting to scientists and clinicians. However, the authors are requested to address the following comments

1- The authors should add one paragraph declaring the clinical usage and its spectrum. 

2- The Section of future direction must be more comprehensive

3- Please revise the typos and grammatical errors.

Author Response

The current study reviews the therapeutic drug monitoring benefits and limitations of colistin use in ICU. The authors shed the light on the chemical structure, analysis and pharmacokinetics of colistin providing its benefits and toxicity. The topic is suitable and interesting to scientists and clinicians. However, the authors are requested to address the following comments

First of all, we would like to thank you for your comments and suggestions on improving the text. All have been fully implemented, and we agree that the information on the spectrum and clinical use of colistin was missing. Thus, the inclusion of these issues has undoubtedly improved the quality of the manuscript.

1- The authors should add one paragraph declaring the clinical usage and its spectrum. 

Accepted: We have mentioned these data at several places in the manuscript – colistin spectrum is specified in the Introduction (see lines 47-51 in the tracked changes version), as well as clinical use according to current guidelines (see lines 52-65), and utility of colistin in clinical practice and resistance in part 4 (see lines 365-382). Moreover, the issue of nebulized colistin was extended (see lines 352-361). We believe that in this way, we provide the data required by the reviewer and better balance the ratio of pharmacological and clinical data.

2- The Section of future direction must be more comprehensive

Accepted: The basic directions for future research on colistin are the same, but we have tried to broaden the context of this research to justify our views better (see lines 527-546).

3- Please revise the typos and grammatical errors.

Accepted: The text was comprehensively revised for typographical and grammatical errors by repeatedly checking the text by all authors, using grammar software, and in collaboratio with native speakers. We believe that there are no other errors in the text.

Reviewer 4 Report

I have reviewed the manuscript “Challenges of colistin use in ICU and therapeutic drug monitoring: a literature review” submitted to “Antibiotics” for possible publication. In this review paper, authors have defined the literature very nicely. I found this work interesting, very nicely written and fit well within the scope of this journal. The manuscript doesn’t have any flaws and it represents the topic very well. I have only few suggestions for authors:

1)     to add the conclusive remarks in the abstract.

2)     To add the mechanism of action of colistin

3)     If possible, draw a figure, which shows the mechanism of action

4)     If possible, authors can add a graph chart which shows the resistance pattern of different bacteria to colistin in recent years.

5)     If possible, the authors can add the region-wise data to see where is higher resistance reported (I mean, Asia, Europe etc.)

I congratulate the authors again.

Thank you!

Author Response

I have reviewed the manuscript “Challenges of colistin use in ICU and therapeutic drug monitoring: a literature review” submitted to “Antibiotics” for possible publication. In this review paper, authors have defined the literature very nicely. I found this work interesting, very nicely written and fit well within the scope of this journal. The manuscript doesn’t have any flaws and it represents the topic very well. I have only few suggestions for authors:

First of all, we would like to thank you for your positive general comments and specific suggestions for improving the text. Two of them were fully accepted, three were only partially incorporated, but we still believe to the reviewer's satisfaction. We agree that information on the mechanism of action and resistance to colistin was lacking. Thus, the inclusion of these issues undoubtedly improved the quality of the manuscript.

1)     to add the conclusive remarks in the abstract.

Accepted: Main gaps and future research directions are mentioned in the abstract now - they summarize the most critical remarks of the review (see lines 23-29 in the tracked changes version).

2)     To add the mechanism of action of colistin

Accepted: Currently proposed mechanisms of action were mentioned in 2.1 Colistin chemical profile, mechanism of action part (see lines 90-105 in the TC version). Several theories of the mechanism of action involve chemical properties, therefore, this topic fits well here.

3)     If possible, draw a figure, which shows the mechanism of action

Partially accepted: We have elaborated more on the mechanism of action of colistin in the text, including a reference to a recent review that graphically summarizes all the mechanisms of action mentioned in the text. A more in-depth discussion of the mechanism of action of colistin in the form of an originally drawn diagram is, from our point of view, beyond the main scope of this manuscript. Our aim in the manuscript was to address the main issues relevant to therapeutic drug monitoring (quantification methods, sample stability, sampling, benefits and limitations for TDM of colistin, including routine use in clinical practice) and related issues of drug stability and drug-material interference.

We believe a reference to the literature with a brief commentary on lines 90-105 will satisfy the reviewer.

4)     If possible, authors can add a graph chart which shows the resistance pattern of different bacteria to colistin in recent years.

Partially accepted: We added several pieces of information on colistin resistance in part 4 (see lines 365-378 in the TC version), including 2 references to the recent reviews on the global prevalence of colistin resistance in CRAB and Klebsiella (ref. 81 and 82; line 369). A deeper analysis of resistance pattern and distribution is beyond the main scope of this manuscript, from our point of view – the scope described above.

We believe that references to the literature with brief comments on lines 365-378 will be satisfactory to the reviewer.

5)     If possible, the authors can add the region-wise data to see where is higher resistance reported (I mean, Asia, Europe etc.)

Partially accepted: We added several pieces of information on colistin resistance in part 4 (see lines 365-378 in the TC version), including 2 references to the recent reviews on the global prevalence of colistin resistance in CRAB and Klebsiella (ref. 81 and 82; line 369). A deeper analysis of resistance pattern and distribution is beyond the main scope of this manuscript, from our point of view – the scope described above.

We believe that references to the literature describing the global prevalence of colistin resistance in two main pathogens, together with brief comments on lines 365-378, will be satisfactory to the reviewer.

Round 2

Reviewer 2 Report

Effectively addressed comments. 

Reviewer 3 Report

The authors responded well to all reviewer raised comments